# Perceptions of Tobacco Price Policy among Students from Sapienza University of Rome: Can This Policy Mitigate Smoking Addiction and Its Health Impacts?

**DOI:** 10.3390/healthcare12090944

**Published:** 2024-05-04

**Authors:** Martina Antinozzi, Susanna Caminada, Mariano Amendola, Vittoria Cammalleri, Barbara Dorelli, Monica Giffi, Felice Giordano, Alessandra Marani, Roberta Noemi Pocino, Davide Renzi, Alessandro Sindoni, Maria Sofia Cattaruzza

**Affiliations:** 1Department of Public Health and Infectious Diseases, Sapienza University of Rome, 00185 Rome, Italy; susanna.caminada@uniroma1.it (S.C.); vittoria.cammalleri@uniroma1.it (V.C.); barbara.dorelli@uniroma1.it (B.D.); monica.giffi@uniroma1.it (M.G.); robertanoemi.pocino@uniroma1.it (R.N.P.); alessandro.sindoni@uniroma1.it (A.S.); mariasofia.cattaruzza@uniroma1.it (M.S.C.); 2Local Health Unit Roma 1, 00168 Rome, Italy; m.amendola@sanita.it; 3National Institute of Health (Istituto Superiore di Sanità), 00162 Roma, Italy; felice.giordano@iss.it; 4Lazzaro Spallanzani National Institute for Infectious Diseases, 00149 Roma, Italy; alessandra.marani@inmi.it; 5District 1, Local Health Unit Roma 1, 00185 Rome, Italy; davide.renzi@aslroma1.it

**Keywords:** smoking, tobacco tax increase, young adults, students, NCDs, tobacco policy, tobacco control, novel tobacco products, smoking cessation

## Abstract

Tobacco use is one of the main risk factors for non-communicable diseases. Avoiding youth initiation and treating addiction are fundamental public health issues to ensure better health. Among tobacco control policies, increasing tobacco price is the single most effective intervention. It reduces tobacco consumption, especially among youths, while representing a government financing source. This study aimed to assess the agreement with the proposal of a one-euro increase in tobacco price earmarked to health issues among students at Sapienza University. Two convenience samples were surveyed, five years apart, on World No Tobacco Days. Smoking habits, agreement with the proposal and reasons for it were collected. Results from the 208 questionnaires (107 in 2014, 101 in 2019) showed 46.6% of agreement with the proposal (53.3% in 2014, 39.2% in 2019, *p* = 0.044). Main predictive factor for agreement was being a non-smoker (OR = 6.33 *p* < 0.001), main reason (64.8%) was it could trigger smokers to quit or reduce consumption. Several factors might have influenced this finding, including the introduction of novel tobacco products and their increased advertisement on social media. In 2024, European Union is planning to update the Tobacco Taxation Directive which could greatly contribute to the reduction of non-communicable diseases and premature deaths.

## 1. Introduction

The tobacco epidemic is a global problem with serious consequences for human health, causing 8 million deaths worldwide every year. Noncommunicable diseases (NCDs) are the major cause of death all over the world, and are caused by genetic and physiological determinants, behavioral risk factors and environmental impacts. Among all smoking-related illnesses, tobacco use plays a fundamental role in the development of noncommunicable diseases [1,2]. 

Youth smoking is a major public health concern, both for its short-term and long-term consequences on health [3]. In Italy, tobacco smoking prevalence among young people is high: in 2018 the highest proportion was registered in the 20–24-year-old age group, both for men (32.4%) and women (22.2%) and in 2022, the last available year from ISTAT, it was 28.0% and 18.9%, respectively [4]. 

The level of tobacco consumption among young people can also be considered as a predictive factor for the development of non-communicable diseases in the future. Specifically, scientific evidence shows that smoking causes a significant distortion of physiological balance, dysregulating the sympathetic nervous system, the renin–angiotensin–aldosterone system and the immune system, as well as leading to a disruption of physiological insulin and oxidant/antioxidant homeostasis, manifested as oxidative stress and insulin resistance, all pathogenetic factors associated with the development of NCDs. Moreover, cigarette smoking has clearly been linked to the development of cardiovascular diseases, chronic obstructive pulmonary disease, hypertension, cancer, as well as many chronic systemic diseases with inflammatory components such as atherosclerosis, Crohn’s disease, rheumatoid arthritis, psoriasis, Graves’ ophthalmopathy and type 2 diabetes [5]. All this evidence highlights the importance, for public health professionals, of reducing tobacco consumption at all ages, in order to improve health and prevent disease development.

According to the World Health Organization (WHO), tobacco taxation may reduce tobacco consumption healthcare costs, while representing a financing source for governments. At the same time, increasing the price of tobacco could be a useful strategy to discourage young people from starting smoking as well as promoting smoking cessation [6,7]. 

For the reasons stated above, most countries worldwide have ratified the WHO Framework Convention on Tobacco (FCTC) to address the global tobacco epidemic, including the prevention of tobacco initiation among young people. In recent years, most tobacco prevention efforts (including policies, age restrictions and media campaigns) have mainly targeted teenagers. Recent studies show that young adults also represent a key target for future action, especially university students [8,9,10].

In this context, to help the implementation of countries’ policies aimed at decreasing tobacco demand, in 2008, the WHO FCTC developed MPOWER. It is a policy package made up of the following six evidence-based policies: monitoring the use of tobacco and available prevention policies; protecting people from tobacco smoke exposure; offering help in tobacco quitting attempts; warning about dangers related to the use of tobacco; enforcing all interventions that ban tobacco advertising and promotion at numerous levels (journals, social networks, advertising signs on the streets…) and sponsorship; and raising the tax burden on tobacco products [6,11]. The subsequent increase in tobacco tax revenues, moreover, could be instrumental in covering expenditures related to tobacco prevention and control programs [12].

In particular, raising taxes on tobacco is considered one of the most cost-effective solutions for reducing tobacco use, especially among young people [13,14]. 

This study aimed to evaluate the agreement of samples from Sapienza University students with a possible one-euro increase in tobacco prices earmarked for health promotion and tobacco cessation actions, considering that this policy has already been highlighted as an important incentive for agreement from previous studies [15,16]. 

Previous Italian studies estimated that a one-euro increase (about 20%) in the price of cigarettes would lead to a 6.8% reduction in smoking prevalence. This would translate into about 750,000 fewer smokers, a reduction in tobacco products consumption for those who continue to smoke, and, consequently, a reduction in exposure to second-hand smoke for non-smokers. The result would be the avoidance of approximately 131,000 disability adjusted life years (DALYs) approximately 15 years after the adoption of the measure [17].

Young people are generally more sensitive to changes in cigarette prices than adults [18,19,20,21]. According to Mannocci et al., this may be due firstly to the addictive nature of cigarette smoking. In particular, younger people who had been smoking for a rather short period are more likely to change their behavior, adjusting more quickly to changes in price compared to adult smokers. Moreover, the amount of money available for young people to use on their personal interests is less than that of adults, and usually comes from the small amount of money given by parents to children from time to time, making the tobacco price increase a possible good deterrent for this particular target population. Furthermore, the behavior of young people is more likely to be influenced by peers compared to adults. Considering all this data, researchers have found that tobacco price increases can lead to a direct and indirect reduction in youth smoking, acting on youth disposable income as well as on peer influence [22,23,24]. Therefore, tobacco taxation policy, together with other public health interventions, such as education, could have a crucial impact on the prevention of tobacco initiation and tobacco cessation for young adults.

Universities potentially play a key role in preventing smoking among students. Progressively, universities around the world are implementing new strategies and public health policies with this aim, such as smoke-free and tobacco-free campuses [25,26]. Sapienza University of Rome, one of the most prestigious Italian Universities, and the largest in Europe (supporting over 115,000 students [27]), is also introducing new policies to become a tobacco-free campus in the next few years, starting with a smoking ban in all university structures, including transit rooms and toilets, in accordance with the rules laid down in the 2008, 2017, 2018 administrative circulars. 

## 2. Materials and Methods

During the World-No-Tobacco-Days (WNTD) [28] in 2014 and 2019, two convenience samples of students from Sapienza University of Rome, Italy were interviewed. The aim was to assess the students’ agreement on the possible introduction of a one-euro tax increase for each pack of cigarettes or tobacco to be spent on prevention. In 2014, the WNTD theme was “Raise taxes on tobacco”, while in 2019 it was “Tobacco and lung health”. 

Each year, on the WNTD, Sapienza University Tobacco Control Unit (UNITAB) [29] organizes informative–formative desks throughout the campus center in popular places for students to pass through. Students are invited to read posters, take self-help materials, and measure breath carbon monoxide (CO) (CO Monitor, Clement Clark International Limited, Mountain Ash, UK).

All students stopping at the desk were asked to fill out an anonymous questionnaire and 100% of them accepted to participate in the study (N = 107 in 2014 and N = 101 in 2019). By agreeing to participate in the study, students gave their consent to the anonymous collection and analysis of data and its publication. 

The questionnaire, constructed ad hoc, was pre-tested by all co-authors and by a sample of students from Sapienza University to verify question language, flow, clarity, readability, and completeness along with the acceptability of the questions and response alternatives. It took approximately 5 min to complete, and it was made up of two parts, as follows: one to be answered by all students, aimed at understanding their attitudes; and one part only for smokers, to better define their smoking behaviors. The questions (multiple choice and open-ended): were grouped into the following 3 sections: (i) personal data (age, gender); (ii) smoking status; and (iii) agreement with the proposed one-euro increase in tobacco price—to be used for health promotion and tobacco cessation actions—and possible reasons for agreement/disagreement (the only open-ended question). Two additional questions were addressed only to smokers: the number of cigarettes smoked per day and the number of smoking-years. Smokers were then considered regular smokers when smoking at least one cigarette per week and occasional smokers when usually smoking less than one cigarette per week [30]. Answers from these two questions were used to calculate pack-years, a value of lifetime tobacco exposure measured as the mean number of cigarettes smoked per day multiplied by the number of years spent smoking, divided by 20 (number of cigarettes per pack). 

Due to the experimental nature and unestablished habits typical of youth, during the analyses, occasional smokers were included in the smokers’ category. 

An additional section of the questionnaire, concerning the use of electronic cigarettes, was administered to all respondents in 2014 but only to a subgroup of people interviewed nearby the library in 2019, because of students’ time availability.

A descriptive statistical analysis of items included in the questionnaire was performed. Responses to the survey’s open-ended question, concerning reasons for agreement/disagreement with the tobacco price increase proposal, were grouped into six categories by the investigators for analysis purposes (three pro tax increase and three against it). Continuous variables were expressed as mean ± standard deviation (SD), whereas categorical variables were expressed as proportions. The comparisons between 2014 and 2019 were performed with a chi-squared test for categorical and dichotomous variables, while for continuous variables, the Student’s t test was used. Multivariable logistic regression models were constructed to identify possible factors independently associated with agreement/disagreement with the proposal and with the reasons for agreement or disagreement. Odds ratios (OR) were used to express regression coefficients with a 95% confidence interval providing data on the relative importance of independent variables for the investigated outcome variable. *p*-values were considered statistically significant when <0.05 and all analyses were performed using STATA 17 (StataCorp LLC, 4905 Lakeway Drive, College Station, TX, USA) [31]. 

## 3. Results

In total, 208 students answered the questionnaires (107 in 2014, 101 in 2019) and their characteristics are reported in Table 1. The mean age of the total sample was 22.5 (±2.2) years and very similar values were observed in 2014 and 2019. Additionally, very similar age ranges were observed in the two periods, with a minimum value of 19 years and a maximum value of 30 and 31 years, respectively, in 2014 and 2019. Female students were 58.8% (respectively, 54.5% in 2014 and 63.3% in 2019, with no statistically significant difference between the two samples). 

Smokers (regular and occasional) were 55.1% in 2014 and 69.3% in 2019 and the increase during this period was statistically significant (*p* = 0.035). In 2019, students had breath CO measured; smokers had a mean value of 12.0 ± 8.8 ppm, while non-smokers had a mean value of 2.2 (±1.6) ppm; and the difference was statistically significant (*p* < 0.001) (Table 1). 

A statistically significant increase from 21.0% in 2014 to 46.2% in 2019 (*p* < 0.001) was observed in the number of students who reported having tried electronic cigarettes (Table 1). 

Almost half of the students (46.6%) reported agreeing with the proposal of a one-euro increase in tobacco price earmarked for health promotion and tobacco cessation actions. This percentage was 53.3% in 2014 and decreased to 39.2% in 2019 (*p* = 0.044).

Considering smoking status, among smokers, about 30% of respondents agreed with the proposal (32.2% in 2014 and 27.9% in 2019), with a difference over time of about 4%, while among non-smokers, 74% agreed (79.2% in 2014 and 65.5% in 2019).

Table 2 shows the characteristics and smoking habits of smokers. From 2014 to 2019 the number of cigarettes smoked per day increased from 8.6 (±5.2) to 9.0 (±5.2), the years of smoking increased from 5.5 (±2.5) to 6.1 (±2.9); and pack-years increased from 2.6 (±2.2) to 3.0 (±2.6), although these differences were not statistically significant. The percentage of students who started smoking before the age of 18 increased from 63.2% to 84.6%; this difference was statistically significant.

Table 3 shows factors associated with agreement with the proposal. After adjusting for the year of the survey, age, and gender, being a non-smoker is a strong statistically significant predictor of agreement (OR = 6.33; *p* < 0.001).

Reasons for agreement and disagreement with relative percentages are reported in Table 4. The majority of students who agreed with the proposal considered it to be useful to decrease the number of smokers and the amount of smoking (64.8%); necessary for health, the environment, and the national economy (28.4%); and a deterrent for not starting (6.8%).

Most of those who disagreed with the proposal reported considering it inconclusive because people will continue to smoke regardless of price and behave according to what they consider “personal freedom” (63.2%); others thought the cost of cigarettes is already high enough and thus the proposal will be a detrimental measure for the personal economy (18.9%); still, others stated that they are smokers, without focusing on a specific reason for disagreement (17.9%).

## 4. Discussion

In our sample, support with the proposal of a possible one-euro increase in tobacco price earmarked for health issues received support by almost half of the participants (more than 50% and about 40% of the students, respectively, in 2014 and 2019). In the literature, there is scant and outdated information about support for tobacco taxation increases among the general population and among young people in particular, while this data could be useful for legislators to implement the single most effective tobacco control intervention [7,11,32,33,34]. 

In 2008, while studying factors associated with individuals’ support for a tobacco tax increase in Germany, analyzing the impact of a two-step tobacco price increase, passed on only partly to the consumer because of the tobacco industry’s intervention on the final price (with an increase lower than +1.16 cents per cigarette in 2002 and +1.26 cents per cigarette in 2004), Hanewinkel et al. found that 39.1% of the general population and 34.2% of people between the ages of 14 to 25 years supported it [35]. 

In addition to the time gap, this age group is not easily comparable with ours because it may be influenced by the presence of very young people who are still minors and who may not yet have consolidated opinions about it, since they cannot legally purchase tobacco. 

We also found that smoking status was independently associated with agreement/disagreement, and that support was higher among non-smokers, in accordance with what was found in the study previously mentioned [35]. In particular, we found that 32.2% and 27.9% of smokers, respectively, in 2014 and 2019, agreed with the proposal, while the percentages among non-smokers were, respectively, 79.2% and 65.5%. In the study by Hanewinkel et al. only 9.6% of smokers and 54.3% of non-smokers supported the increase in tobacco taxation. The difference between smokers and non-smokers in supporting such a measure is expected because of the nature of the public health policy, which exclusively effects smokers. In contrast, nonsmokers may be in favor of a tax that they are not required to pay but that could reduce smoking, exposure to secondhand smoke, and the costs of smoking, by which they are directly or indirectly affected. 

The slight decrease in the level of agreement with the possibility of tobacco price rises among Sapienza University students could be related to a decreased awareness of the effectiveness of tobacco taxation. Indeed, in 2014 the World No Tobacco Day Campaign theme was about the effect of raising tobacco taxes, while in 2019 the focus was on lung health. Moreover, the introduction on the Italian market of new types of e-cigarettes and novel tobacco products, advertised on social media and public places, could have played a key role in “normalizing” the act of smoking.

Even if coming from a small (not generalizable) and younger sample, our findings are in line with those found in Germany and could be very useful to help filling the gap of evidence on this topic.

In Italy, between 2014 and 2019, there was a slight increase (about 14%) in the price of cigarettes, from an average price of about EUR 4.50 to EUR 5.10 per cigarette pack (considering 15 popular brands). Our proposal of a one-euro increase would therefore mean about a 20% increase in the average price of a cigarette pack. 

Almost 65% of students stated that a one-euro increase in tobacco price may trigger smokers to re-evaluate the financial burden of smoking, consider the opportunity to quit and take action. This is particularly important because young people generally think they can quit anytime, without difficulties, and thus they postpone their cessation. Increasing tobacco prices could be a great opportunity for those having low intention to quit to urge them to take action [32,33,34,35]. In 2000, a Canadian study aimed at evaluating attitudes toward tobacco control, the authors reported that 51.6% of non-smokers and 27.2% of smokers were seen to “strongly agree” or “agree” with higher taxes on tobacco because they would help people to quit smoking [36].

Some other students pointed out that an increase in tobacco prices could lead to a major burden on low-income people, forcing them to buy cheaper products, with a possible higher health risk (e.g., cheaper quality of cigarettes’ components or substances used to make cigarettes by the tobacco company). This is consistent with evidence from the scientific literature [19,37,38] and with the WHO Report on Tobacco Inequities, which states that it should be mandatory, alongside any price increase, to ensure affordable and accessible nicotine replacement therapy (NRT) and smoking cessation support to low-income groups, so that this taxation policy does not become a source of inequality [39]. It is important also to consider that it is the poorest population that benefits most from an increase in tobacco taxation, as they smoke more, are more susceptible to increases, and suffer more from adverse health consequences. Interventions that reduce the prevalence of tobacco use help to reduce exposure and health inequalities [40].

One risk related to a tobacco taxation increase only on cigarettes can be the switch of smokers to novel tobacco products to avoid the taxation. To prevent this issue and to be effective, increased taxation must cover all products and all countries, otherwise inequality in tax rates could facilitate tobacco consumption. 

Italy, following the example of other EU member states, prohibited e-cigarette advertising on any platforms as well as cross-border advertising and sponsorship, ratifying Legislative Decree 6 on 12 January 2016. However, “indirect advertising” is still very common, for “vaping” shops specialized in novel tobacco products (e.g., advertising signs or products displayed on shop windows that are not subjected to any kind of restrictions), as well as for heated tobacco products advertising on-site and online. Recently, the Tobacco Industry has been subject to sanctions by the Italian Antitrust Authorities for misleading advertising of heated tobacco products (HTP) sold as simple electronic devices [41]. Moreover, the implementation of definite provisions contained in the Legislative Decree is still unclear, such as the activities related to e-cig production and distribution monitoring that should be performed by the Italian Ministry of Health [42,43].

In this context, the introduction of novel tobacco products could be considered as a confounding factor in the rate of agreements on the tobacco price increase. However, this factor was not a big issue at the moment we collected data, as at that time, the spread of novel tobacco products on the market was just beginning [44]. At the same time, the Tobacco Industry started targeting young people as perfect buyers for their new products on the market, using influencers on social media for advertisement, sometimes through subliminal advertising [45,46,47,48]. 

Considering these marketing techniques, it seems important to underline that creating smoking prevention policies addressed particularly to young people, even on social media [49], should be the focus of all governments in the years to come, to prevent addiction in the new generations and to avoid an increase in younger customers for tobacco industries [23,50,51,52].

This study has some limitations that should be considered when interpreting the results. First, convenience sampling might not be representative of the whole university population. One of the weaknesses of the questionnaire was that it did not investigate the respondents’ faculty, thus did not allow for the answers to be adjusted according to the students’ backgrounds. Moreover, it was not possible to collect information about the socioeconomic status of respondents, because of the sensitive nature of this kind of data. However, it is important to note that Vardavas et al., in their study, reported that support for an increase in tobacco taxation was not associated with socioeconomic status and income [15]. 

Other limitations are the small number of participants and the high percentage of smokers in our sample. Indeed, smokers could have been more interested in stopping at our desk than non-smokers, due to the topic we investigated. Moreover, we considered regular and occasional smokers who at this age are quite a lot due to the experimental phase typical of youths. Also, in Italy, in recent years, tobacco control measures have weakened due to increasing interference from the Tobacco Industry (proposals to increase tobacco taxation are systematically rejected and new tobacco company plants have been officially inaugurated by representatives of State) [53]. All these circumstances could have led to a selection bias.

The use of a self-administered questionnaire could also have resulted in possible response biases in participants’ answers. 

Conversely, the strengths of this study are that, to our knowledge, this is the first study specifically investigating young adults’ opinions on a tobacco tax increase, as well as using a biomarker (breath CO) to assess smoking status in 2019, whereas most similar studies examined public opinion in general. 

This study interviewed students at the same university campus, comparable by age and gender, five years apart, and thus offers the possibility to formulate hypotheses about the perceptions of the problem among students over time, considering the possible influence of the WNTD themes on students’ opinions. Indeed, in 2014, the World No Tobacco Day Campaign theme was about the effect of raising tobacco taxes, while in 2019, the focus was on lung health. 

Further research (including qualitative studies) is needed to verify and understand reasons for supporting a tobacco price increase. A better understanding of the students’ points of view may contribute to more effective policies aimed at contrasting smoking in young adults. 

## 5. Conclusions

The European Union plans to create, by 2040, a tobacco-free generation in which less than 5% of the European population consumes tobacco products (European Cancer Plan). Various actions and regulatory measures, including updating the Tobacco Taxation Directive (TTD) are under consideration. The directive is expected to increase tax rates and harmonize the tax treatment of tobacco products in the 27 countries. 

According to a Smoke Free Partnership forecast, if the TTD is revised in 2024 and the reforms start operating in 2025, by year 2028, 1,089,325 and 93,615 premature deaths will be averted, respectively, in Europe and Italy [54].

Furthermore, even if risk perceptions of smoking-related health issues seems to have been decreasing over time [55], public health professionals should take into account the importance of tobacco use as one of the main risk factors in the NCDs’ development, considering the burden of smokers (active or passive ones) as future patients. Nowadays, the struggle is not only about WNTD but, following the path of Big Tobacco’s marketing strategies, involvement in this battle against tobacco smoking should become a daily effort. As long as tobacco advertisement is funded mostly through product placement because of a public advertising law ban, public health policies should target smokers’ everyday lives, e.g., through posts and videos on social media or TV advertising. 

This small cross-sectional study is important as a starting point off the beaten track of popular tobacco cessation research; investigating young people’s opinions is important in the long run, as youth will be the adult population not long from now, so they are the perfect targets for new tobacco cessation campaigns to aim at. Investing in youth smoking cessation results in better health for future adults, meaning that less money is spent on tobacco-induced diseases in the future and that, overall, less lives will be wasted because of tobacco consumption. Young people should be the main target of upcoming research, especially considering their potential as present and future customers of Big Tobacco and, consequently, of national health care systems.

## Figures and Tables

**Table 1 healthcare-12-00944-t001:** Sample’s characteristics and agreement with the one-euro increase proposal.

	All (N = 208)	2014 (N = 107)	2019 (N = 101)	*p*-Value
Age	22.5 ± 2.2	22.8 ± 2.6	22.1 ± 2.9	0.076
Gender	N = 199	N = 101	N = 98	
Male	82 (41.2%)	46 (45.5%)	36 (37.7%)	0.207
Female	117 (58.8%)	55 (54.5%)	62 (63.3%)	
Smoking status	N = 208	N = 107	N = 101	
Smoker	129 (62.0%)	59 (55.1%)	70 (69.3%)	**0.035**
Non-smoker	79 (38.0%)	48 (44.9%)	31 (30.7%)	
CO level				
Smokers			12.0 ± 8.8 ppm	**0.001**
Non-smokers			2.2 ± 1.6 ppm	
E-cigarette	N = 157	N = 105	N = 52	
Tried at least once	46 (29.3%)	22 (21.0%)	24 (46.2%)	**<0.001**
Never tried	111 (70.7%)	83 (79.0%)	28 (53.8%)	
One-euro tax increase proposal	N = 204	N = 107	N = 97	
Total agreement	95 (46.6%)	57 (53.3%)	38 (39.2%)	**0.044**
Smokers	38 (29.9%)	19 (32.2%)	19 (27.9%)	0.601
Non-smokers	57 (74.0%)	38 (79.2%)	19 (65.5%)	0.186

Values in bold are statistically significant.

**Table 2 healthcare-12-00944-t002:** Characteristics and smoking habits of smokers.

	All (N = 129)	2014 (N = 59)	2019 (N = 70)	*p*-Value
Cigarettes/day	8.8 ± 5.2	8.6 ± 5.2	9.0 ± 5.2	0.731
Years of smoke	5.8 ± 2.7	5.5 ± 2.5	6.1 ± 2.9	0.186
Pack-years	2.8 ± 2.5	2.6 ± 2.2	3 ± 2.6	0.499
History of smoking (>5 s)	61 (50%)	28 (49.1%)	33 (50.8%)	0.904
Initiation age				
<18	91 (74.6%)	36 (63.2%)	55 (84.6%)	**0.007**
≥18	31 (25.4%)	21 (36.8%)	10 (15.4%)	

Values in bold are statistically significant.

**Table 3 healthcare-12-00944-t003:** Logistic regression model regarding the overall agreement with the proposal of a one-euro tax increase.

	Overall Agreement OR (95%CI)	*p*-Value
Year		
2014	Ref.	
2019	0.70 (0.37–1.31)	0.260
Age (continuous)	1.03 (0.92–1.16)	0.573
Gender		
Male	Ref.	
Female	1.40 (0.73–2.70)	0.309
Smoking status		
Smoker	Ref.	
Non-smoker	6.33 (3.25–12.34)	**<0.001**

Values in bold are statistically significant.

**Table 4 healthcare-12-00944-t004:** Reasons associated with the agreement or disagreement with the one-euro increase in the tobacco tax proposal.

Agree Because		Disagree Because	
Useful for smoking cessation and/or reduction in consumption	64.8%	It will be an inconclusive measure	63.2%
Necessary for health, environment and national economy	28.4%	The cost is already too high	18.9%
It is a deterrent for smoking initiation	6.8%	I am a smoker	17.9%

## Data Availability

Based on the consent form indicating that the data will not be shared publicly for confidentiality, the datasets generated and analyzed during the current study are not publicly available but are available from the corresponding author on reasonable request.

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
