# Peer review of "Perceptions of Tobacco Price Policy among Students from Sapienza University of Rome: Can This Policy Mitigate Smoking Addiction and Its Health Impacts?"

_healthcare, 2024, doi:10.3390/healthcare12090944_

Round 1
Reviewer 1 Report
Comments and Suggestions for Authors
The title is very general; it is not connected with the objective. Title should generate interest about the study. Example you can include phases sucha as’ youth/students perceptions of tobacco price policy’,
Introduction:
After background of the study, begin by explicitly stating the research objectives and hypotheses. Clearly articulate what the study aims to investigate and what specific questions it seeks to answer. This will provide readers with a better understanding of the study’s purpose.
Method:
Elaborate on the representativeness of the student samples from Sapienza University. Consider whether the findings can be generalized beyond this specific population.
Provide more details about the survey instrument, data collection process, and response rate.
Discussion:
Address the decrease in agreement with the proposal from 2014 to 2019. Explore potential reasons for this decline. Were there changes in public perception, media influence, or other external factors?
References such as 1,3,4 25,28,29,30 are incomplete.
Author Response
REVIEWER 1
Comments and Suggestions for Authors
Q. The title is very general; it is not connected with the objective. Title should generate interest about the study. Example you can include phases sucha as’ youth/students perceptions of tobacco price policy’,
A. Thank you for the suggestion, we decided to change the previous title: “Proposal of tobacco control policy to prevent smoking-related non-communicable diseases: preliminary results among samples of students from Sapienza University of Rome.” to “Perceptions of tobacco price policy among students from Sapienza University of Rome: can this policy mitigate smoking addiction and its health impact?” to make the sense of the study clearer and to generate interest about the study.
Q. Introduction: After background of the study, begin by explicitly stating the research objectives and hypotheses. Clearly articulate what the study aims to investigate and what specific questions it seeks to answer. This will provide readers with a better understanding of the study’s purpose.
A. We thank you for this advice, we tried to change the structure of introduction, explicitly stating research objectives in lines 85-88, to make it easier for reader to understand study’s purpose.
Q. Method: Elaborate on the representativeness of the student samples from Sapienza University. Consider whether the findings can be generalized beyond this specific population. Provide more details about the survey instrument, data collection process, and response rate.
A. Thank you for the suggestion, we added some more details on the survey instrument and data collection (lines 135-146). Due to the small sample these results are not generalizable beyond this specific population (lines 266-268). Information on response rate were added (lines 131-132).
Q. Discussion: Address the decrease in agreement with the proposal from 2014 to 2019. Explore potential reasons for this decline. Were there changes in public perception, media influence, or other external factors?
A. We thank you for the suggestion, we addressed the decrease in agreement in lines 259-265.
Q. References such as 1,3,4 25,28,29,30 are incomplete.
A. Thank you for the observation, we fixed the references.

Reviewer 2 Report
Comments and Suggestions for Authors
Overall, the paper is well written. I have some minor revision suggestions.
1. Page 2 of 12
Line 83-85.
"In Italy, tobacco smoking prevalence among young people is high; in 2018 the highest proportion was registered in the 20-24-year-old age group, both for men (32.4%) and women (22.2%)"
The author may consider using more recent references regarding tobacco smoking prevalence in Italy, if possible.
2. Page 3 of 12
Line: 129
"The questionnaire, constructed ad hoc, was pre-tested by all co-authors"
Authors may consider uploading the questionnaire as supplementary material for readers to have a better understanding of the questions.
3. Page 3 of 12
Line : 138
"questions were addressed only to smokers (both regular and occasional)"
It is recommended for authors to add details on the definition of regular and occasional smokers used in the study.
4. Page 6 of 12
Line number: 225
"In 2008, in Germany, Hanewinkel et al. studying factors associated with individuals’ support on tobacco tax increase"
Authors may need to provide more details or background of this study in Germany, such as the amount of tobacco tax increases.
Author Response
REVIEWER 2
Comments and Suggestions for Authors
Overall, the paper is well written. I have some minor revision suggestions.
1. Page 2 of 12 Line 83-85. "In Italy, tobacco smoking prevalence among young people is high; in 2018 the highest proportion was registered in the 20-24-year-old age group, both for men (32.4%) and women (22.2%)" The author may consider using more recent references regarding tobacco smoking prevalence in Italy, if possible.
A. Thank you for your suggestion, we added also more recent reference regarding tobacco smoking prevalence in Italy (lines 47-48)
2. Page 3 of 12 Line: 129 "The questionnaire, constructed ad hoc, was pre-tested by all co-authors" Authors may consider uploading the questionnaire as supplementary material for readers to have a better understanding of the questions.
A. Thank you for the advice. However, having detailed the composition of questionnaires in the article main text (lines 140-150) and considering that they are written only in Italian, we decided not to upload the questionnaires as supplementary material.
3. Page 3 of 12 Line : 138 "questions were addressed only to smokers (both regular and occasional)" It is recommended for authors to add details on the definition of regular and occasional smokers used in the study.
A. Thank you for the suggestion, we added details (+reference) on the definition of regular and occasional smokers in lines 145-147.
4. Page 6 of 12 Line number: 225 "In 2008, in Germany, Hanewinkel et al. studying factors associated with individuals’ support on tobacco tax increase" Authors may need to provide more details or background of this study in Germany, such as the amount of tobacco tax increases.
A. We thank you for this observation, we provided more details on this study in Germany in lines 238-243.
